# Detecting nematic order in STM/STS data
# with artificial intelligence

**Jeremy B. Goetz[1,2], Yi Zhang[2,3] and Michael J. Lawler[1,2⋆]**

**1** Department of Physics, Applied Physic and Astronomy,
Binghamton University, Binghamton, NY, 13902, USA
**2** Laboratory of Atomic And Solid State Physics, Cornell University, Ithaca, NY 14853, USA
**3** International Center for Quantum Materials, Peking University, Beijing, 100871, China

⋆ mlawler@binghamton.edu

## Abstract

Detecting the subtle yet phase defining features in Scanning Tunneling Microscopy and Spectroscopy data remains an important challenge in quantum materials. We meet the challenge of detecting nematic order from the local density of states data with supervised machine learning and artificial neural networks for the difficult scenario without sharp features such as visible lattice Bragg peaks or Friedel oscillation signatures in the Fourier transform spectrum. We train the artificial neural networks to classify simulated data of symmetric and nematic two-dimensional metals in the presence of disorder. The supervised machine learning succeeds only with at least one hidden layer in the ANN architecture, demonstrating it is a higher level of complexity than a nematic order detected from Bragg peaks, which requires just two neurons. We apply the finalized ANN to experimental STM data on $CaFe_2As_2$, and it predicts nematic symmetry breaking with dominating confidence, in agreement with previous analysis. Our results suggest ANNs could be a useful tool for the detection of nematic order in STM data and a variety of other forms of symmetry breaking.



## Contents

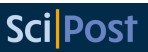
## 1 Introduction

Scanning Tunneling Microscopy (STM) and spectroscopy (STS) data is difficult to fit to theory. These experiments can achieve visualization of the surface electronic structure with atomic level spatial resolution. They do so by bringing a metal tip near the sample surface to allow electron quantum tunneling under a bias voltage. The resulting tunneling current is a function of tip position and applied voltage. From it, the local density of states (LDOS) of the sample is measured and can be compared to the simulations for model interpretation following the solid-state theory. For instance, the impurity scattering of electrons on the surface of a metal may result in a standing wave pattern commonly referred to as quasi-particle interference that depends on the momentum transfer across the Fermi surface [1, 2]. Therefore, we can map out the electronic Fermi surface and the underlying systematic phase and symmetries by interpreting the Fourier transform of the quasi-particle interference pattern. Some other electronic properties, such as the presence of a spectral gap [3], also have smoking gun features. However, it is sometimes hard to connect STM experimental data to idealized models and tangible theories. For example, inhomogeneous behavior found in strongly correlated materials can lead to complex interference and render single-impurity quasi-particle interference analysis irrelevant. Also, the uncertainty, noise, and resolution in measurement are usually hard to account for and lay substantial difficulty towards a smoking-gun judgment for an unbiased theoretical match, especially during close comparisons between facing-off hypothetical models.

Recently, machine learning techniques have seen widespread adoption and increasing utility in the fields of condensed matter physics as a new route for data analysis and model building [4, 5]. Machine learning is a branch of artificial intelligence where systems can learn from data, identify patterns, and make decisions with minimal human intervention. These capacities are consistent with various routine goals and challenges in condensed matter physics - connecting detailed microscopic models with qualitative universal features. Indeed, after training on simulated data from diverse microscopic models categorized into a series of classes following respective hypothetical claims, artificial neural networks (ANNs) can extract information from 'big' STM experimental data, and determine the characteristic symmetries of the realistic electronic quantum matter [6].

The recent trend of applying machine learning techniques to condensed matter physics, beginning with its use in density function theory [7–10] and its extension to strongly correlated electrons models [5, 11] suggests a new route to extracting information from STM data [6, 12]. Specifically, one can train artificial intelligence (AI) architecture such as ANNs on simulated data from diverse microscopic models to capture a macroscopic phase defining feature of inter-

est. Then we can ask this AI for its judgment on a realistic data set that may or may not exhibit this phase defining feature. By using simulated data sets with rich and detailed microscopic information, the ANN can extract the feature even when it manifests differently under different microscopic settings. Intriguingly, much like following a renormalization group flow, through machine learning, the ANN summarizes the relevant phase defining features automatically [4].

With this in mind, consider the case of detecting nematic order in STM data. Nematic order describes the onset of discrete anisotropy that breaks systematic four-fold rotation symmetry $C_4$ down to two-fold rotation symmetry $C_2$. Detecting nematic order in STM or STS data can sometimes become a challenge [13]. One origin of the challenge is instrumental, as an anisotropic output could originate from that of the STM metal tip instead of the sample. For example, the claims of nematic orders in $Bi_2Sr_2CaCu_2O_{8+x}$ following analysis of Bragg peaks [13–15] were questioned [16] until evidence of nematic domains was later discovered [17]. Another difficulty arises in the absence of sharp features such as Bragg peaks, which have helped to establish the presence of nematic order in $CaFe_2As_2$ [18,19]. Further, a large amount of disorder, poor spatial resolution, and limited field of view also add to the complications. Even though we can sometimes feel ambiguously that the LDOS pattern 'seems nematic' when such anisotropy is strong, a quantitative analysis is lacking in general.

Here we revisit the challenge of detecting nematic order in the absence of sharp features from a machine-learning perspective. We choose simulated STM images representing the LDOS of various tight-binding models on a two-dimensional square lattice in the presence of various types of impurities as our training sets. To provide a range of dispersion and Fermi surfaces, we also vary the hopping amplitudes in the tight-binding models, which we categorize into two classes according to the presence or absence of the global four-fold rotation symmetry. We limit the number of sublattice sites and LDOS pixels to one per unit cell. Thus, there are no meaningful Bragg peaks. The Friedel oscillation signatures are also lost when the density of impurities is too large, or system size is too small, and traditional analysis fails to identify the nematic order. In comparison, using supervised machine learning, we succeed in training ANN architecture with one hidden layer with sufficient neurons to distinguish the two symmetry classes with high accuracy. On the contrary, we analytically show that only two hidden layer neurons are necessary if the data supports Bragg peaks and can be distinguished by Fourier transform. Finally, we input the realistic STM data of $CaFe_2As_2$ [19] into a successfully finalized ANN, and the ANN output suggests a nematic symmetry breaking with dominating confidence, in support of the claims in Ref. [19]. Remarkably, the experimental data set has a longer variation length scale compared to the simulated data set. Therefore, our results further demonstrate the utility of ANNs for STM data analysis and their capacity to capture phase-defining universal physics from abundant microscopic information. We note that by adding an additional category and corresponding training set devoted to an anisotropic metal tip scenario, it may be possible to train the ANN to distinguish the microscopic source of the anisotropy and eliminate the potential instrumental bias as well. However, this is beyond the scope of the current paper and left for future work.

The rest of the paper is organized as follows: in the next section, we present our model setup for the simulated LDOS data, the architecture of ANN, and the supervised machine learning algorithm. Further, we discuss the results of training and compare our approach with traditional ones. In Sec. III, we study the application of our ANN on the STM experimental data in $CaFe_2As_2$. Sec. IV is our conclusion and future outlook. Several appendices support the methods and results presented in section II.

## 2 Method and Results

### 2.1 Models and data set generation

As a simple model of a nematic ordered material, we consider a tight-binding model with anisotropic hopping on the square lattice

$$H_0 = -\sum_{\mathbf{r},\alpha} t_\alpha c^\dagger_{\mathbf{r}+\alpha} c_{\mathbf{r}} + h.c. - \mu \sum_{\mathbf{r}} c^\dagger_{\mathbf{r}} c_{\mathbf{r}}, \tag{1}$$

where $\mathbf{r} + \alpha$ denotes one of the nearest neighbor sites to the site $\mathbf{r}$. At the chemical potential $\mu$, this model roughly emulates the expected normal-state band structure of an over-doped cuprate superconductor. Further, the hopping parameters $t_\alpha$ characterize the spatial symmetry, and the difference between the horizontal bond and the vertical bond introduces the nematic order. The specific choices of model parameters are presented in Appendix A.

For our purposes, however, this model is too simple for non-trivial local density of states (LDOS) $N_{\mathbf{r}}(\omega)$ (defined below), since it is invariant under translation and scalar under rotation - spatially isotropic even with hopping $t_x \neq t_y$. In reality, there are further complications due to the sub-unit-cell structure factors and impurities, which generate more information as well as challenges. Here, we focus on the latter and add to the Hamiltonian $H = H_0 + H_{imp}$ the following terms

$$H_{imp} = \sum_{\mathbf{r},\alpha} \delta t_{\mathbf{r}\alpha} c^\dagger_{\mathbf{r}+\alpha} c_{\mathbf{r}} + h.c. + \sum_{\mathbf{r}} \delta \mu_{\mathbf{r}} c^\dagger_{\mathbf{r}} c_{\mathbf{r}}, \tag{2}$$

where $\delta t_{\mathbf{r}\alpha}$ and $\delta \mu_{\mathbf{r}}$ are only finite at a few locations and characterize the strength of on-bond and on-site local quenched disorders, respectively. The settings of the random disorders, including its density, distribution, etc. are also presented in Appendix A.

For a given Hamiltonian $H$, we compute $N_{\mathbf{r}}(\omega)$ via the imaginary part of the Green's function

$$N_{\mathbf{r}}(\omega) = -2\mathrm{Im}\tilde{G}_{\mathbf{r},\mathbf{r}}(\omega), \quad G_{\mathbf{r},\mathbf{r}'}(\omega) = \langle r | \frac{1}{\omega - H + i\epsilon} | r' \rangle, \tag{3}$$

where $\epsilon = 10^{-5}$ is a small imaginary part characterizing the width of the energy level. The frequency $\omega$ can be absorbed into the chemical potential $\mu$ and is neglected afterward. Using different parameters for hopping, chemical potential and impurities, we generate about 6,000 images of $N_{\mathbf{r}}(\omega)$ for the nematic data set ($t_x \neq t_y$) as well as for the symmetric data set ($t_x = t_y$ and so on for the other $t_\alpha$'s). Presumably, the resulting data set contains the diverse types of impurity generated anisotropy in both the nematic and the symmetric systems. In the following, we attempt to coarse grain the 'big,' detailed data in our data sets and summarize the essence of nematic order using ANN-based AI method, and then generalize its application to realistic experimental data beyond the simulations.

### 2.2 Artificial neural network presentation of nematic order parameter

A fully-connected, feed-forward ANN consists of sequential layers of neurons. Each neuron processes the inputs from all the neurons from the previous layer according to the associated parameters known as the weights $\mathbf{W}$ and the bias $b$, the threshold above which a neuron fires, and outputs to the trailing neurons:

$$y = \sigma(\mathbf{W} \cdot \mathbf{x} + \mathbf{b}), \tag{4}$$

where $\mathbf{x}$ is the input and $\sigma$ is the sigmoid function $\sigma(x) = 1/(e^{-x} + 1)$. Due to the existing and efficient optimization algorithm, this architecture and its descendants are highly popular

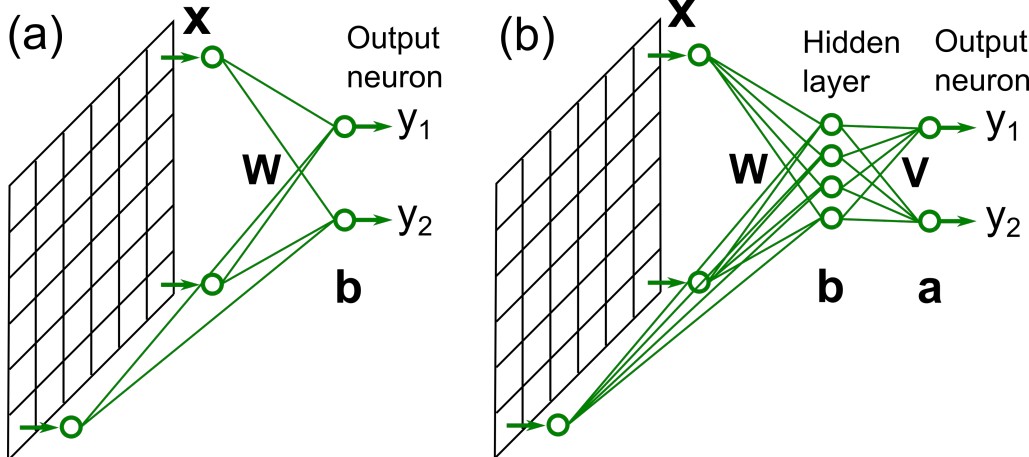

Figure 1: ANNs with the LDOS $N_{\mathbf{r}}(\omega)$ as the inputs $\mathbf{x}$ give normalized outputs $y_1$ and $y_2$ as the ANNs' probabilistic judgment and confidence on the input image being symmetric or nematic, respectively. Each neuron processes its inputs $\mathbf{x}'$ and outputs as $y' = \sigma(\mathbf{W} \cdot \mathbf{x}' + \mathbf{b})$, where $\mathbf{W}$ are the weights associated with each of the inputs, $b$ is a bias, and $\sigma$ is a sigmoid function. (a) An ANN with no hidden layer, and (b) an ANN with one hidden layer are examples of the fully-connected feed-forward ANNs.

for AI applications. For instance, Fig. 1(a) illustrates a simple architecture for an ANN with no hidden layer.

Let's consider the conventional measures of the nematic order from an ANN perspective. Conventionally [13–15], a nematic order parameter

$$O_N = \sum_{\mathbf{r}} N_{\mathbf{r}} \left[ \cos(\mathbf{Q}_x \cdot \mathbf{r}) - \cos(\mathbf{Q}_y \cdot \mathbf{r}) \right] \tag{5}$$

can be defined, where $\mathbf{Q}_x = (2\pi/a, 0)$ and $\mathbf{Q}_y = (0, 2\pi/a)$ are the wave vectors of the two Bragg peaks related by a 90-deg rotation. Essentially, such treatment is a Fourier transform of the STM image, and linear in the input data $N_{\mathbf{r}}$. So we can interpret this as an ANN with a hidden layer of just two neurons, one to detect if $O_N > 0$ and the other to detect if $O_N < 0$. This is achieved by setting $\mathbf{x} = \mathbf{N}_{\mathbf{r}}$, $b = 0$, and $\mathbf{W} = \cos(\mathbf{Q}_x \cdot \mathbf{r}) - \cos(\mathbf{Q}_y \cdot \mathbf{r})$ so that the output of the first hidden layer neuron is $y_1 = \sigma(O_N)$. The weight of the second hidden layer neuron is $-\mathbf{W}$ for the output $y_2 = \sigma(-O_N)$, and we declare an image nematic if either neuron fires: the output of the two hidden layer neurons is fed to two output neurons, one acting as an "or" gate that fires if either $y_1$ or $y_2$ is positive and the other a "nor" gate, which does the opposite. Of course, we do not take the $O_N$ as directly meaningful, but instead, compare it to a noise floor. These neurons are therefore equivalent to the sigmoid function $0 \leq \sigma(\pm O_N - |b|) \leq 1$ where $|b|$ is the value of $|O_N|$ in a noisy isotropic image. Therefore, we conclude that the problem of detecting nematic order in the presence of Bragg peaks can be captured by an ANN with just two hidden layer neurons.

In the absence of Bragg peaks, the situation appears more complicated. In this case, the disorder is necessary to observe anisotropy. However, the disorder may also serve as a curse. A local impurity can scatter electrons and generate Friedel oscillations, which spread out anisotropically (isotropically) away from the impurity in an anisotropic (isotropic) system. Signatures of these oscillations are present in the Fourier transform of the LDOS $N_{\mathbf{r}}(\omega)$. However, complications arise when the field of view is small, or the density of impurities is large. In this case, the route through direct analysis of Friedel oscillations breaks down (see pros and cons details of Friedel oscillations as a measure of anisotropy in Appendix B).

## 2.3 Artificial neural network with no hidden layer

In contrast to conventional approaches that study Friedel oscillations or Bragg peaks, our method is based upon the search for an ANN capable of identifying anisotropy in a data set, notably in hard data set where Bragg peaks and clear Friedel oscillation signatures are absent, and conventional approaches struggle. We start with the warm-up ANN with no hidden layer, and then we show that a single hidden layer with multiple neurons is indeed necessary to detect nematic order.

We use both the Tensor Flow package from Google and custom routines for ANN calculations. We have 256 input neurons representing the LDOS $N_r$ at each of the sites on the $16 \times 16$ lattice. We also have two output neurons with normalized outputs $y_1 + y_2 = 1$. The outputs $y_1$ and $y_2$ represent ANN's probabilistic judgment and confidence for the nematic order and the symmetric phase for the model system behind the input $\mathbf{x} = \mathbf{N_r}$, respectively. We also normalize $\mathbf{x}$ to ensure input consistency over diverse model systems. The corresponding weight $\mathbf{W}$ is a $256 \times 2$ matrix, and the bias $b$ is a $1 \times 2$ vector. Fig. 1(a) shows an example of such an ANN. We use a supervised machine learning algorithm to training the ANN and optimize the weights and biases using the gradient descent method together with back propagation so that the outputs are as consistent as the known nematic and symmetric classifications of the training data sets as possible(see Appendix C for details).

Notably, the ANN with no hidden layer only obtains a maximal 53% accuracy and is hardly better than coin-flipping for classifying the data sets. The residue of the cross-entropy cost function (see appendix C) remains high, indicating that there exist inconsistencies between the ANN predictions and the correct classifications. Further, there is little improvement upon prolonged training and an increased number of epochs (iterations through the data set).

This failure is not surprising. We note that detecting order is non-monotonic in the order parameter ($|O_N|^2 > 0$), and the neural network with no hidden layer is limited to monotonic expressibility and regressions [20]. Indeed, our discussion of $O_N$ above suggests that we need two hidden neurons to detect nematic order in the presence of Bragg peaks. In the following, we use an ANN architecture with one hidden layer and capable of non-monotonic expressibility and re-examine the supervised machine learning on the same data sets.

## 2.4 Artificial neural network with one hidden layer

Fig. 1(b) is an illustration of an ANN with a single hidden layer. In practice, our ANN's hidden layer consists of 31 sigmoid neurons, which are fully connected to the input neurons through the $256 \times 31$ weight matrix $\mathbf{W}$. Similarly, the output neurons are fully connected to the hidden layer neurons through the $31 \times 2$ weight matrix $\mathbf{V}$. We also introduce a $1 \times 31$ vector $b$ and $1 \times 2$ vector $a$ as the biases for the hidden layer neurons and output neurons, respectively. We address further details on the single-hidden-layer ANN architecture and supervised machine learning algorithm in Appendix C.

With the ANN with a single hidden layer, we can achieve a satisfactory over 95 % accuracy straightforwardly. Tuning of the hyper-parameters such as the learning rate, regularization, and the number of hidden layer neurons, can further bring down the loss and increase the accuracy to a stage near 100 % accuracy. In the following, we focus on an ANN with a more generally reachable 95 % accuracy as a sufficiently good demonstration of classification and to avoid potential over-fitting issues.

To visualize the power of the ANN, we plot in figure 2 two typical images from our database that is labeled nematic (hopping $t_x \neq t_y$) and symmetric (hopping $t_x = t_y$), respectively. Articulating the universal differences between the images as different phases is difficult for us. On the other hand, the trained ANN has successfully incorporated these subtle rules, thus not only labels the images correctly but also with relatively high confidence: most of the nematic

sample images have $y_1 > 0.95$ and most of the symmetric sample images have $y_2 > 0.95$, see Fig. 2.

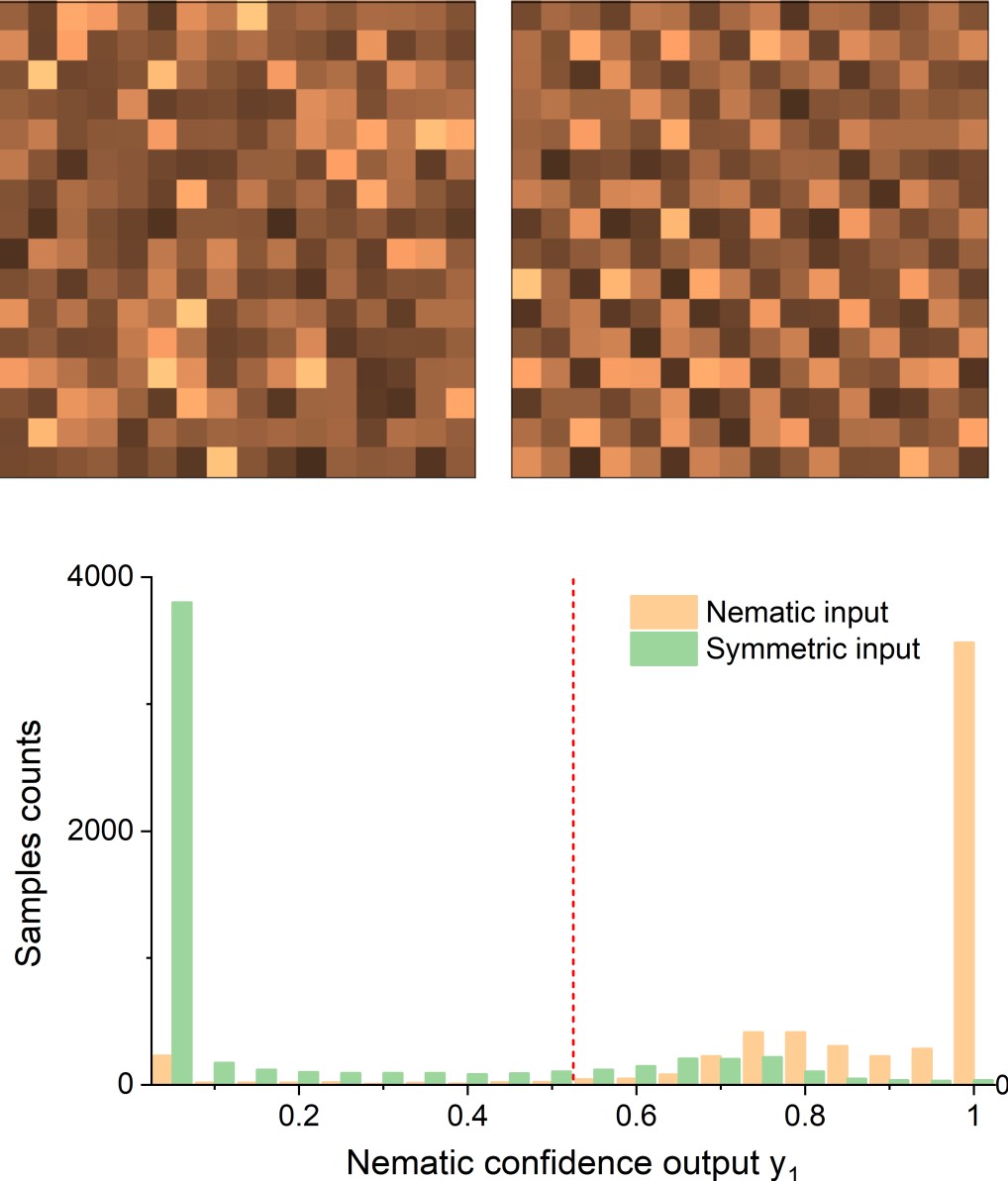

Figure 2: (Left) A simulated LDOS $N(r)$ image for the nematic phase, and (right) a simulated LDOS $N(r)$ image for the symmetric phase. The finalized ANN with a single hidden layer correctly identifies the classification with over 95% confidence for both images. While we may observe elusive traces of anisotropy between the two images with our naked eyes, the signatures are nevertheless hard to summarize and quantify for a clear-cut detection. (Bottom) the distribution of the neuron output $y_1$ among 5940 nematic sample images and 5940 symmetric sample images. The ANN confidence of the symmetric phase $y_2 = 1 - y_1$. For most of the sample images, the ANN is able to distinguish the correct phase with high confidence.

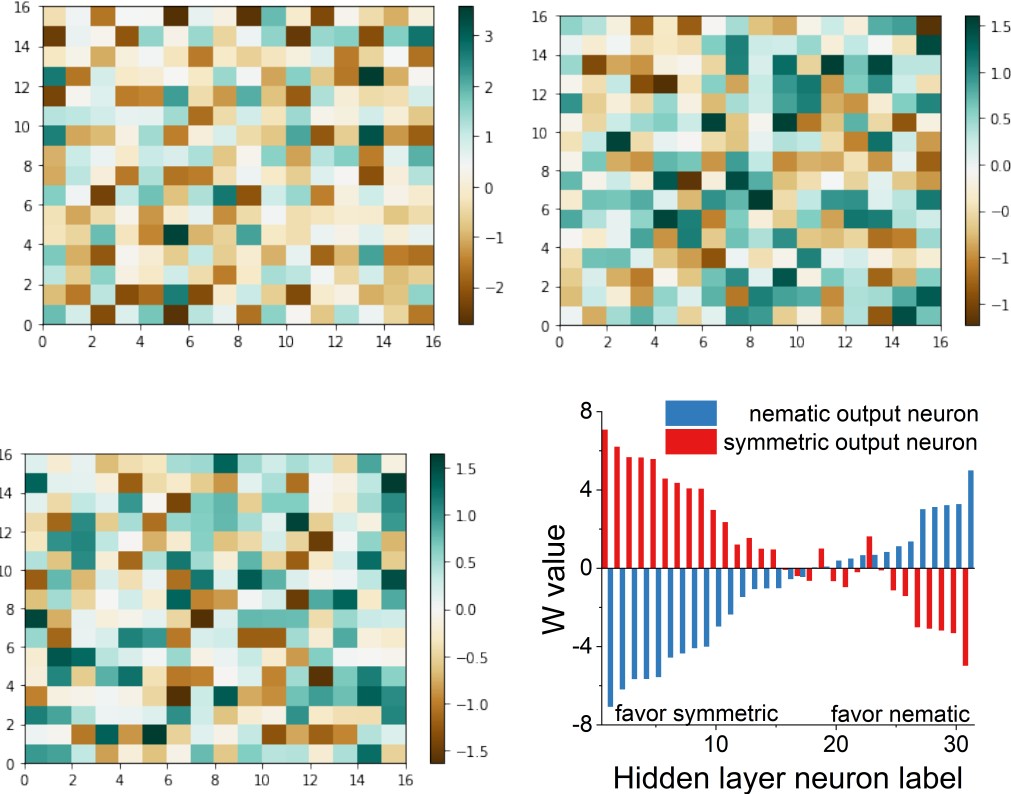

Figure 3: To analyze the criteria the ANN with a hidden layer detects the nematic order, we visualize the weights **W** between the 16 × 16 input neurons and the hidden layer neuron (top left) with the largest contribution to the output neuron for the symmetric phase, and (top right) largest contribution to the output neuron for the nematic phase, as well as (bottom left) relatively neutral contributions to both the symmetric and nematic phases. It seems that identifying nematic order in these images depicting firing condition is as difficult a case as in the original STM images. (Bottom right) the distribution of the weights between the hidden layer neurons and the output neurons. The ANN outputs involve the collective contributions from multiple neurons in the hidden layer.

## 2.5 ANN sensitivity and criteria for nematic order

Both the lack of Bragg's peaks and the failure of ANN with no hidden layer imply the detection of nematic order is likely associated with higher-order terms of the $N(\mathbf{r})$ inputs, e.g., the correlations. To go one step deeper, we analyze which correlations the ANN is detecting and whether such correlations are related to the nematic symmetry breaking.

As an oversimplified version of the interactions between the neurons in an ANN, each neuron in the hidden layer weighs the pixels of and search for features in an STM image independently, and then the output neurons weigh the hidden neurons and establish correlations between selected features that contribute to the decision output, e.g., nematic or symmetric. These features are the key. Based on this reasoning, we present in Fig. 3 visualizations of the weights used by a hidden neuron in assessing a simulated STM image input. The hidden neurons we present are one that contributes the most to the symmetric-phase output, one that contributes the most to the nematic-phase output, and one that contributes little to the decision. The distinction between them and even those original STM images of the nematic phase (Fig. 2(left)) is fairly elusive and lacks smoking-gun characteristics, at least to our eyes.

Therefore, understanding the criteria for nematic order from the ANN parameters remains a problem no less challenging than understanding the hidden traits of the nematic order itself. Still, our results suggest that the criteria are unlikely simply linear or monotonic in $N(\mathbf{r})$, or it would be dominated by fewer hidden neurons, rather than the interplay between hidden-layer neurons and thus correlations between the features and patterns. Though not direct evidence, this also implies that the ANN does not resort to some direct, spurious signals but instead looks for subtle nematic signatures that are hard to visualize.

## 3  ANN application to experimental STM data

In Refs. [18,19], nematic anisotropy was observed in STM data on $CaFe_2As_2$, an iron-based superconductor, in several ways. One of the analyses revealing apparent nematicity via Fourier transform [18], also known as quasi-particle interference, is similar to our example in Appendix B. Also, the distinction between correlation lengths along different directions is in line with the nematic symmetry breaking [19]. These references concluded that the data appears like 'leaves fallen randomly on the ground, yet with all the leaves pointing in the same direction,' as is seen in Figure 4. The anisotropy is not easy to confirm by our naked eyes and to make matters worse, the absence of positional order means the lack of Bragg peaks.

To seek additional evidence for this proposed nematic anisotropy, we pass the realistic STM data on $CaFe_2As_2$ to our trained AI architecture - the ANN with one hidden layer. We pass numerous $16 \times 16$ images randomly cut from Ref. [19], similar to the top panel inset in Fig. 4, to the ANN and assess the statistics of the outputs. We note that these tested experimental images look very different from the simulated ones used for training the ANN, and are much smoother on the pixel scale containing only longer wavelength information. Also, we one-to-one replace the sorted values in the experimental image samples with those in the simulated ones. In this way, we keep the topography of the image while allowing a consistent LDOS value distribution and thus better compatibility of the experimental data with the ANN. A similar treatment is to renormalize the experimental data so that its average and standard deviation are consistent with the simulated ones. The conclusions that follow do not qualitatively depend on the selected treatment or the chosen simulated image sample for the replacement LDOS values.

Interestingly, over 1000 experimental sample images, the ANN claims an overwhelming 90% possess a nematic order ($y_1 > 0.5$), and $> 53\%$ of the samples with high confidence $y_1 > 0.8$. Further, we sample the pixels with $2 \times 2$, $3 \times 3$, $5 \times 5$ intervals, etc., essentially coarse-graining larger plaquettes with a single representing pixel within. The ANN remains confident in the nematicity of these images, yet the confidence gradually wanes, and the outputs asymptotically approach a coin-flip as the scaling increases. We summarize the data in Fig. 4. Therefore, it is likely that our ANN can generalize beyond the simulations and the specific mechanism used to create the training set, and relies on more generic properties of the nematicity that readily transfers to longer-scale scenarios - effectively follows a renormalization group flow. We note that similar behaviors are also observed for the tight-binding-model data with different coarse-graining scales, see Appendix D for details. The ANN gets confused when the scales differ too much, which can be alleviated by an expansion of the training set to add simulations with more diverse length scales, either initially or recursively. We leave such attempts for enhanced adaptability to future work.

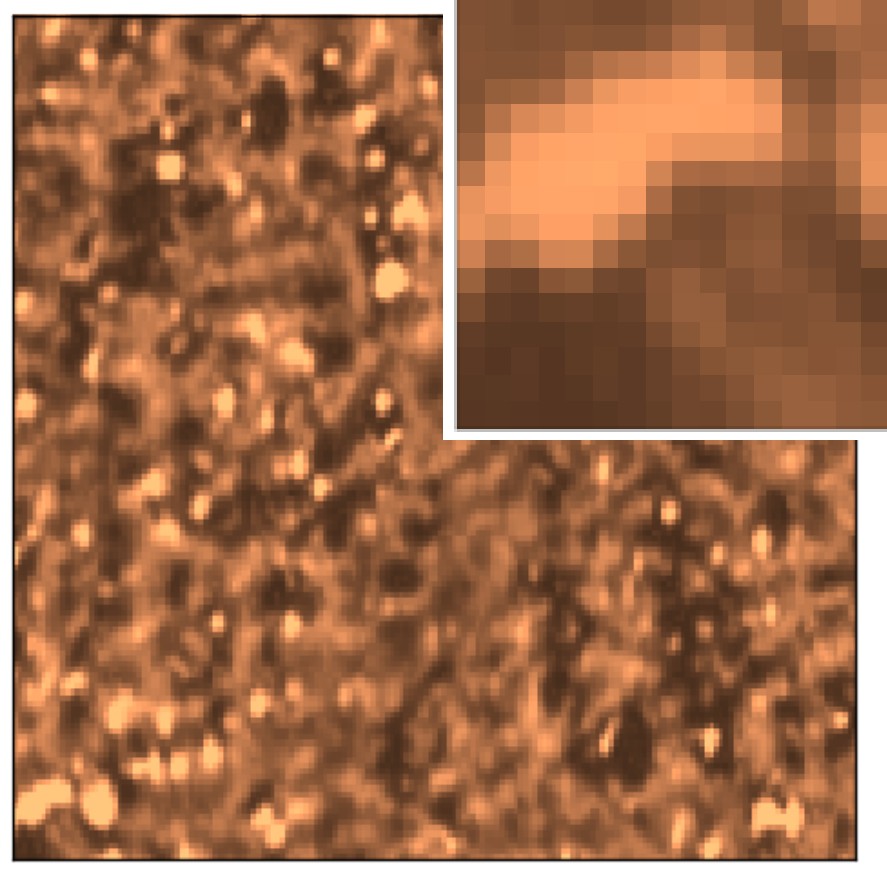

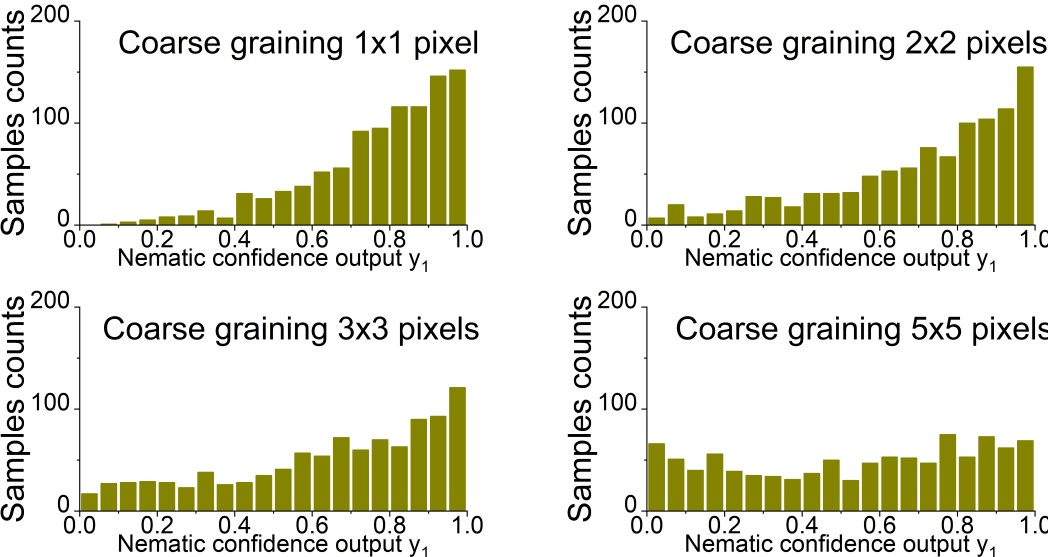

Figure 4: (Top) Data taken from Ref. [19] Supplementary materials fig. S3f (same as Fig. 2b of Ref. [19]) and converted to a monochrome color scale. (Top inset) A typical 16x16 image cut of the data consistent with the input format of the ANN. We take various images as such across the field of view. (Bottom) the distribution of the neuron output $y_1$ as ANN confidence of the nematic phase among 1000 sample images randomly selected across the field of view. The neural outputs show clear confidence for a nematic phase, which gradually wane as the coarse-graining scale increases.

# 4 Conclusion and Outlook

We have trained ANNs with either no hidden layer or one hidden layer via a supervised machine learning algorithm to detect the presence of nematic order in materials based upon the simulated LDOS data sets. Only AI architecture with a hidden layer can successfully distinguish the nematic order from a symmetric phase, likely relying on nontrivial correlations in the data. Remarkably, we find that the ANN not only is more sensitive than the human eye at identifying the nematic order, but also capture hidden, universal defining feature sufficiently even in the absence of Bragg's peaks and clear-cut Friedel oscillation signatures. Finally, we apply our ANN architecture to realistic STM data and obtain an anisotropic response, consistent with the previous consensus. That the results are relatively robust for samples coarse-grained at different length scales implies our ANN can potentially follow a renormalization group flow [21, 22] to the global, universal properties of the nematic order starting from microscopic considerations.

For more rigorous and practical applications in the future, the machine learning training set should contain more generic models, including various interacting models and non-interacting models with broader settings than those we have used for illustration. Also, while the ANN can disclose hidden information in moderately disordered systems (e.g., see Appendix B), they are still helpless and confused in the strong disorder limit (see Appendix D), where the distinctions between the different phases entirely vanish. Consistency checks between multiple samples and fields of view can help to build up confidence and avoid potential misleading mistakes.

# Acknowledgements

We thank Seamus Davis and Eun-Ah Kim for the illuminating discussions. MJL acknowledges support in part by the National Science Foundation under Grant No. NSF PHY-1125915. YZ acknowledges supports from the Bethe fellowship at Cornell University and the start-up grant at International Center for Quantum Materials, Peking University. MJL and YZ acknowledge the hospitality from KITP at the preliminary stages of the work.

# A Generating a database of simulated STM images for nematic and symmetric phases

We generate a database of simulated STM images from spinless fermion systems on the square lattice as discussed in section IIA of the main text. The model systems are characterized by the following parameters:

$t_\alpha$: The uniform hopping strength in the $\alpha \in \{x, y, x+y, -x+y, 2x, 2y\}$-directions are drawn from the ranges $t_x, t_y \in [1.1, 1.5]$, $t_{x+y}, t_{-x+y} \in [0.6, 0.8]$, $t_{2x} = t_{2y} = 0.3$.

$\mu$: The chemical potential sets the filling of the Fermi sea. We set $\mu \in [0.95, 1.05]$ (very over-doped for a normal-state cuprate model).

$\delta t_{\mathbf{r}\alpha}$: $\delta t_{\mathbf{r}\alpha}$ represents a local distortion of the otherwise-uniform hopping between a randomly chosen site $\mathbf{r}$ and a neighboring site $\mathbf{r} + \alpha$. We set $\delta t_{\mathbf{r}x} = \delta t_{\mathbf{r}y} = 0.15$, $\delta t_{\mathbf{r}x+y} = \delta t_{\mathbf{r}-x+y} = 0.1$ and $\delta t_{\mathbf{r}2x} = \delta t_{\mathbf{r}2y} = 0.05$.

$\delta \mu_{\mathbf{r}}$: $\delta \mu_{\mathbf{r}} = 0.2$ represents a local quenched disorder as a chemical potential change on randomly chosen site site $\mathbf{r}$.

$\omega$: $\omega$ defines the energy of the states the STM tunnels electrons into relative the chemical potential $\mu$. In practice, we neglect $\omega$ and absorb it into $\mu$.

All of these parameters are randomly varied over the ranges discussed above to generate 5940 simulated LDOS images, 5400 of which are used as the training set and the rest for validating purposes. A typical image has between 8 and 52 impurities, which account for 3% to 20% impurity concentrations.

Finally, we present the Fermi surface and band structure for a typical model in the absence of impurities in Fig. 5. For the set of models with the range of parameters mentioned above, the band structures do not change much more than the line width in the plots.

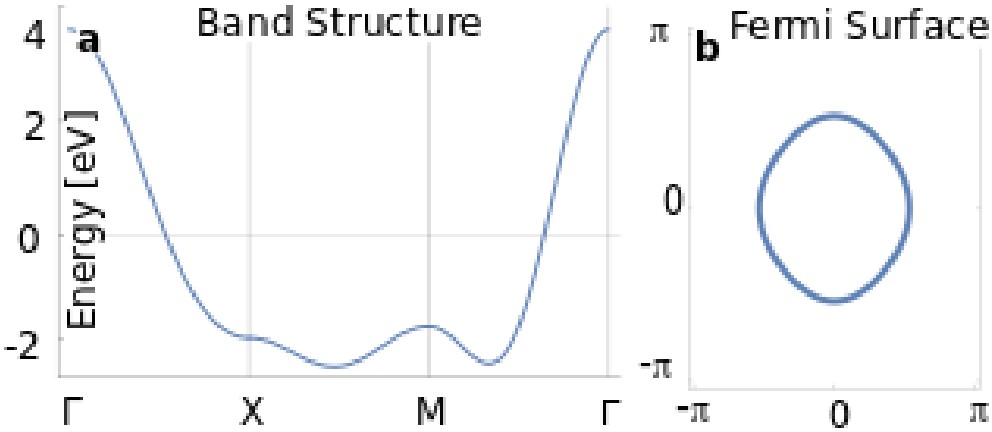

Figure 5: Typical band structure and Fermi surface of the clean model system behind the simulated data set described in the main text and this appendix.

# B Comparison with quasi-particle interference techniques

As a comparison with the machine learning approach, we study in this appendix the possibility of identifying the nematic order through conventional methods on the local density of states (LDOS) in metallic systems. The problem is complicated by local impurities, which inevitably break the symmetries of the original pristine system. On the other hand, the presence of impurities allows us to focus on the quasi-particle interference behaviors in the Fourier transform of the LDOS, which is detectable with STM. A comparison between the peak structures along the high symmetry directions along $\hat{x}$ and $\hat{y}$ allows us to determine whether the symmetry connecting them is present or broken by the nematic order. In the following, we discuss the scenarios where this method fails to yield conclusive results, and the introduction of the machine learning approach becomes indeed constructive.

For concreteness, we consider a tight-binding model on a two-dimensional square lattice:

$$
\begin{aligned}
H &= H_0 + H_{imp} \\
H_0 &= -\sum_{\mathbf{r}} t_x c^\dagger_{\mathbf{r}+\hat{x}} c_{\mathbf{r}} + t_y c^\dagger_{\mathbf{r}+\hat{y}} c_{\mathbf{r}} + \text{h.c.} \\
H_{imp} &= \sum_{\mathbf{r} \in R_w} w_{\mathbf{r}} c^\dagger_{\mathbf{r}} c_{\mathbf{r}},
\end{aligned}
\tag{6}
$$

where $R_w$ is a set of positions with quenched disorders and $w_{\mathbf{r}} \in [-W, W]$ is the on-site disorder strength. We also study a system of size $L \times L$ with periodic boundary conditions in both the

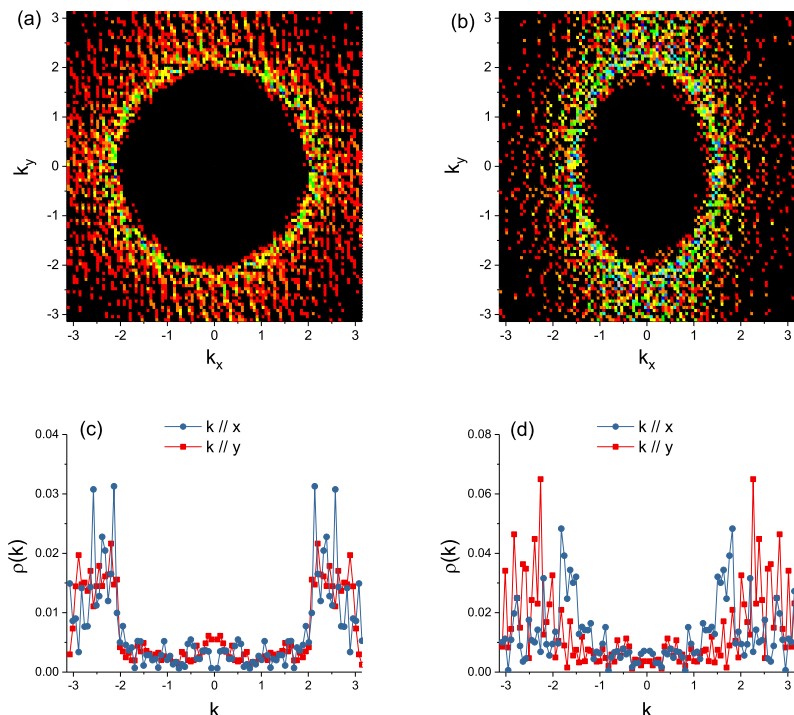

Figure 6: The Fourier transform $\rho(k)$ of the LDOS for (a) a symmetric model with $t_x = t_y$, and (b) a nematic model with $t_x \neq t_y$. There are 20 random impurities in the $100 \times 100$ system. The contour of the $\rho(\mathbf{k})$ peaks follow closely the geometry of the Fermi surface, thus indicates the model symmetry and the presence or absence of the nematic order. (c) The peak locations of $\rho(\mathbf{k})$ in (a) along the high symmetry directions $\hat{x}$ and $\hat{y}$ overlap, while (d) the peaks of $\rho(\mathbf{k})$ in (b) along the high symmetry directions $\hat{x}$ and $\hat{y}$ clearly sit at different values of $k$.

.

$\hat{x}$ and $\hat{y}$ directions. The corresponding LDOS is obtainable through the Green's function:

$$\rho(\mathbf{r}) = -\frac{1}{\pi}\mathrm{Im}G(\mathbf{r}) \tag{7}$$
$$G = (\mu + i\delta - H)^{-1},$$

where $\delta = 0.01$ is a small imaginary part that introduces a finite width for each energy level. We set $t_x = 1.0$, $t_y = 0.5$, $\mu = -2.5$ for a nematic metal and $t_x = t_y = 1.0$, $\mu = -3.0$ for a symmetric metal. $W = 2.0$. The LDOS is then Fourier transformed into the momentum space for the amplitude $\rho(\mathbf{k})$ at each wave vector.

In the presence of a single impurity (within $L \times L$ lattice sites) and a large system size $L = 100$, the quasi-particle interference pattern has a clear connection to the Fermi surface and the symmetry of the model. This even holds with a moderate amount of impurities, see Fig. 6. A straightforward comparison of the peak locations along the high symmetry $\hat{x}$ and $\hat{y}$ directions reveals whether these two directions are physically equivalent, see Fig. 6(c) and (d). On a $100 \times 100$ system, the sharp $\rho(\mathbf{k})$ behaviors at $\sim 2k_F$ persists even in the presence of 250 disorders, or an equivalent occupancy of 2.5% of the total sites.

Unfortunately, there exist scenarios where this approach fails to meet our goal: (1) when the field of view is limited, which in turn results in sloppy resolution after the Fourier transform, thus making identifying any difference or discrepancy in $k$ difficult; also, when the impurity

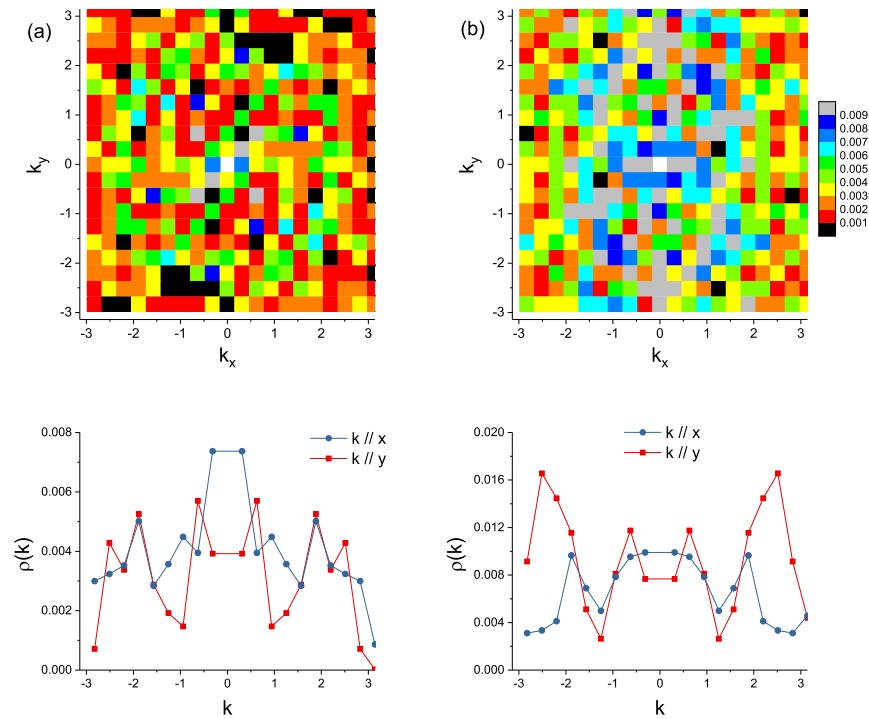

Figure 7: The LDOS Fourier transform $\rho(\mathbf{k})$ profile of (a) a symmetric model, and (b) a nematic model. The models are identical to those in Fig. 6. Yet there are 40 impurities on a smaller $20 \times 20$ system. (c) and (d) are $\rho(\mathbf{k})$ along the $\hat{x}$ and $\hat{y}$ directions.

density becomes overly large, the Fourier transform becomes too noisy to convey any useful information. As examples, we show in Fig. 7 results of $\rho(k)$ for both the nematic and the symmetric models with a larger density of impurities and a smaller field of view $L \times L$, $L = 20$. Smoking-gun signatures as those in Fig. 6 are no longer available for a clear-cut judgement. Even in the cases where there may exist vague signatures that we trace back to from the model wave vectors, such as Fig. 7(c) and (d), it is difficult to isolate them from the other noisy peaks, especially when we do not have the answer in advance. Likewise, we apply Fourier transform to the sample data sets we used for machine learning, and a meaningful, interpretable sharp peak signature in $\rho(k)$ is absent in general. One solution to increase the signal-noise ratio is to average over disorder configurations, see Fig. 8; however, a large amount of LDOS data set of the same sample or model is necessary to make this approach available.

Therefore, we conclude that the application of Fourier transform on the LDOS data is only helpful in determining nematic order when we have a sufficient field of view with relatively sparse impurities, while the machine learning approach focusing on the original LDOS is not limited in these scenarios. Another scenario where this Fermi-surface-sensitive scheme fails is when the broken symmetry is in the Fermi velocities instead of the Fermi vectors at the specific Fermi energy. For instance, consider the following model Hamiltonian:

$$H_0 = -\sum_{\mathbf{r}} t_x c^{\dagger}_{\mathbf{r}+\hat{x}} c_{\mathbf{r}} + t_y c^{\dagger}_{\mathbf{r}+\hat{y}} c_{\mathbf{r}} + \tilde{t}_y c^{\dagger}_{\mathbf{r}+2\hat{y}} c_{\mathbf{r}} + \text{h.c.},$$

(8)

where we set $t_x = 1.0$, $t_y = 0.5$, $\tilde{t}_y = 0.167$. The model is clearly nematic. On the other hand, the Fermi surface given by the dispersion $\epsilon_{\mathbf{k}} = -2\cos(k_x) - \cos(k_y) - 0.334\cos(2k_y)$ is

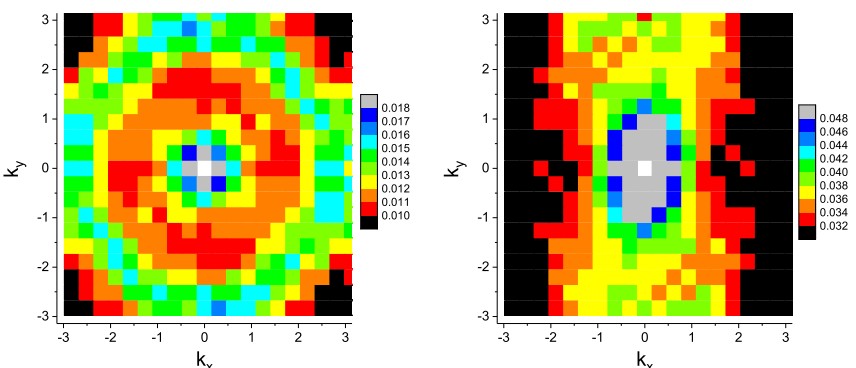

Figure 8: The LDOS Fourier transform $\rho(\mathbf{k})$ profile averaged over 1000 disorder configurations for (a) a symmetric model, and (b) a nematic model. The rest of the settings are identical to those in Fig. 7.

close to isotropic at $\mu = -2.333$, see Fig. 9(a). This leads to the absence of nematic behaviors in the Fourier transform $\rho(\mathbf{k})$ of the LDOS, see Fig. 9(c) and (d). In comparison, machine learning approaches are based upon the original real-space LDOS data and thus may look beyond merely the Fermi wave vectors.

## C  Supervised machine learning algorithm for training ANNs

We train our ANNs using stochastic gradient descent and back propagation algorithms, commonly used in supervised machine learning, to minimize the cross-entropy cost function that characterizes the distance between the expected output $y^*$ and the actual ANN output $y$, e.g. the predicted nematic order confidence of an image and the actual presence or absence of the nematic order:

$$S = \sum_{\mathbf{x},i} y_i^* \log[y_i(\mathbf{x})], \tag{9}$$

where the summation is over all samples $\mathbf{x}$ and output neurons $i$. We also add a small L2 regularization to the cost function to reduce over-fitting.

In each training epoch, we divide the entire training set into smaller batches stochastically, evaluate each batch's cost function partial derivative with respect to each one of the weights $W_j$ and biases $b_k$ via back propagation, and modify the weights and biases following a step size called learning rate $\eta$:

$$\begin{aligned} \Delta W_j &= W_j - \eta \cdot \partial S / \partial W_j \\ \Delta b_k &= b_k - \eta \cdot \partial S / \partial b_k. \end{aligned} \tag{10}$$

In practice, we start with a relatively large learning rate for efficiency and lower $\eta$ every 100 epochs to allow for better convergence. Our batch size is 50 images. After each epoch, we measure the accuracy of a separate validation set of 540 images to determine the convergence and over-fitting conditions. Also, we constantly vary the hyper-parameters such as learning rate, L2 regularization constant, batch size, number of neurons in the hidden layer, etc. for better results.

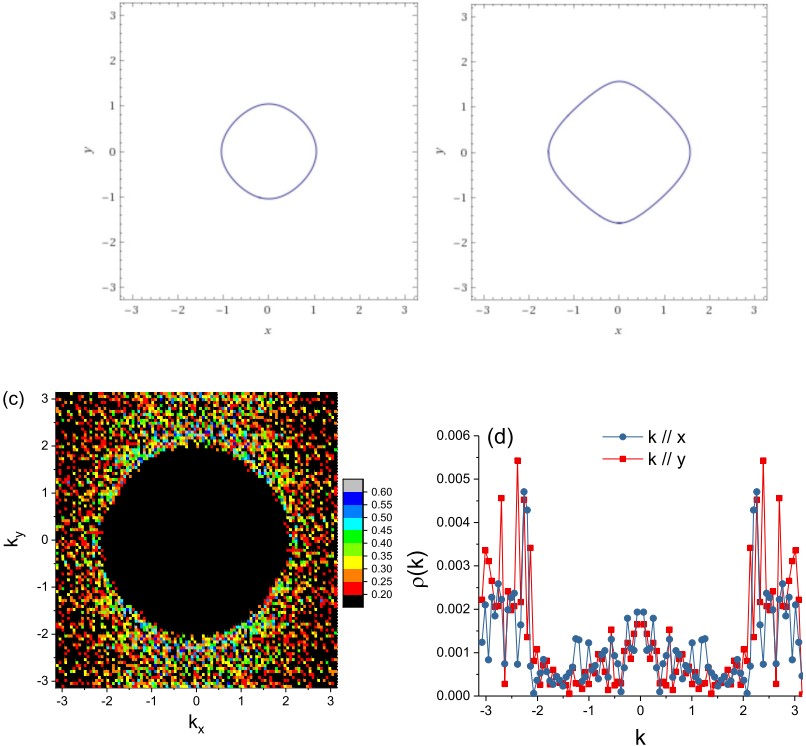

Figure 9: (a) The Fermi surface of the model in Eq. 8 with parameters $t_x = 1.0$, $t_y = 0.5$, $\tilde{t}_y = 0.167$, $\mu = -2.333$ leads to a nearly symmetric Fermi surface. (b) Another example with parameters $t_x = 1.0$, $t_y = 0.5$, $\tilde{t}_y = 0.25$ and $\mu = -1.5$ shows a nearly symmetric Fermi surface. (c) The Fourier transform of the LDOS $\rho(\mathbf{k})$ and (d) $\rho(\mathbf{k})$ along the high symmetry directions of (b) seem rather consistent with a symmetric phase despite the model in Eq. 8 is explicitly nematic.

## D   ANN test results for additional scenarios

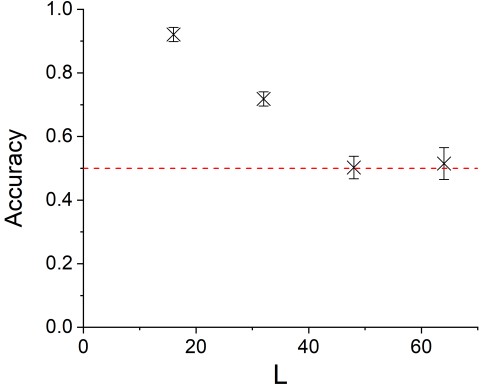

Figure 10: The accuracy of the ANN output decreases when the coarse-graining scale significantly differs from the training set. The parameters of the tight-binding models are identical to those in the main text and summarized in Appendix A and the system sizes are $L \times L$ with $L = 16, 32, 48, 64$, respectively. The LDOS data is then reduced to a $16 \times 16$ field of view and, in turn, assessed by the ANN.

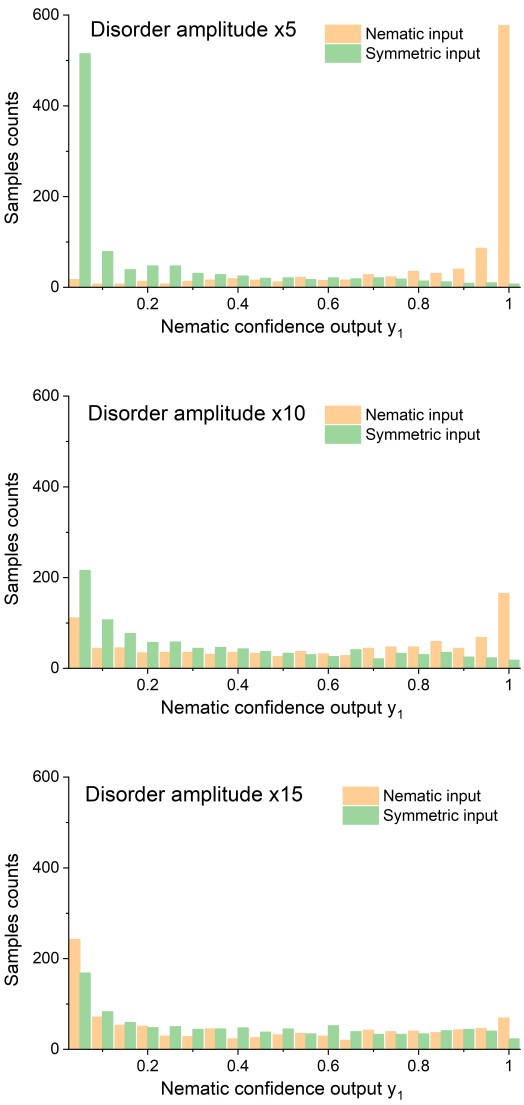

Figure 11: The accuracy of the ANN output decreases as the system approaches the strong disorder limit. The setting is similar to the lower panel in Fig. 2, yet with an impurity density around the upper limit and a much stronger $\delta t_{\mathbf{r}\alpha}$ and $\delta\mu_{\mathbf{r}}$ amplitudes. The ANN accuracy is no better than a coin flip 50% in the last case.

In this appendix, we show the ANN results for additional cases of the tight-binding model to explore certain boundaries and limitations of the method outlined in the main text. First, we examine the impact of scaling and coarse graining on the accuracy of the ANN outputs: we generate LDOS data for system sizes $L \times L$, $L = 16, 32, 48, 64$, and select the pixels with intervals, so that the inputting fields of view all take the size of $16 \times 16$. We have tested 2000 samples for $L = 16$ and $L = 32$, 1000 samples for $L = 48$, and 200 samples for $L = 64$, respectively. The corresponding accuracy of the ANN output is summarized in Fig. 10.

Next, we study the ANN performance in the presence of stronger disorder, where the different phases increasingly become physically indistinguishable. Correspondingly, we retain the tight-binding model parameters and $16 \times 16$ system size in Appendix A while using the upper limit of the impurity density and increase the disorder strength $\delta t_{\mathbf{r}\alpha}$ and $\delta\mu_{\mathbf{r}}$ by 5 times, 10 times, and 15 times, respectively. The results are summarized in Fig. 11.

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
