# Peer review of "Detecting Nematic Order in STM/STS Data with Artificial Intelligence"

_SciPost Physics, doi:SciPost Phys. 8, 087 (2020)_

## Round 1 · Referee Report · Anonymous · 2019-4-8

Report

The manuscript by Goetz et al. reports on detecting subtle nematic order from local density of states (LDOS) data measured by scanning tunneling microscopy (STM) using supervised machine learning. They simulate LDOS data using various tight binding methods to train and test their artificial neural network (ANN). They obtain 95% accuracy score with an ANN with a single hidden layer. They test the ANN on a real STM data that is observed to be anisotropic in previous studies. The ANN is able to identify nematic order with 99% confidence.

The manuscript is well written, the subject is novel and the results are compelling in my opinion.
However, the authors test the ANN only on a single anisotropic STM image. I would like to see its performance on few other isotropic as well as anisotropic STM images. The fact that the ANN performs poorly on simulated data (only %65 Fig. 2) which it is trained on compared to the real STM data (%99) is a bit controversial.

Typo:
For example, inhomogeneous behavior found in strongly correlated materials [can lead] to ...

---

## Round 2 · Referee Report · Anonymous (Referee 2) · 2020-2-16

Strengths

1 - well written
2 - a useful result with a clear answer for experimental data; approach is generic

Weaknesses

1 - see below under requested changes 2- see below under requested changes 3- see below under requested changes -- this amounts to the fact that the ANN output is difficult to interpret for humans 4 - not entirely clear to me whether the modelling by a non-interacting model is sufficient in case no Bragg peaks or Friedel oscillations are observed.

Report

In "Detecting nematic order in STM/STS data with artificial intelligence" the authors want to enhance the classification of experimentally observed STM images through the use of artificial intelligence.

The problem setting is clear. The authors want do distinguish symmetric from nematic data, or more precisely, data that has $C_4$ symmetry vs data that has $C_2$ symmetry. Detecting nematic order can sometimes become a challenge because of instrumental reasons or large amounts of disorder. To this end the authors train an artificial neural network containing one hidden layer and two output neurons with simulated STM images generated from non-interacting fermionic models. They then let the machine decide on realistic STM data of CaFe2As2 and find a nematic symmetry with confidence. The method is particularly useful in cases where there are no Bragg peaks nor Friedel oscillations.

The non-interacting fermionic models, which can be isotropic or anisotropic, are supplemented with a disorder Hamiltonian. The input parameter is the local density of states. In a first step, the authors show that two neurons are insufficient to detect nematic order (in case of Bragg peaks), ie that a hidden layer is necessary. Next they show that one hidden layer is also sufficient. Finally, they apply the machine to the experimental data.

The paper is well written and easy to follow. The analysis of the ANN appears to be correct.

Requested changes

I have a couple of questions that must be answered prior to any decision on publication.

(1) Upon coarse-graining the pixels the authors write that the confidence of the ANN gradually wanes. Is this also seen in the data from the non-interacting model? Or does one have to include disorder that changes over a scale of a dozen lattice sites to reproduce this?

(2) One of the main conclusions of this work is that the ANN remains successful in distinguishing nematic from symmetric samples even when standard approaches fail, say in case of strong disorde. This is remarkable. Imagine a sample with very strong disorder. Then, and this is a typical argument for disordered systems, it is impossible to say whether this sample was generated from phase A or from phase B in case the disorder distribution is generic (in which case one can continuously change the parameters of the disorder distribution, and hence generate the sample from either phase with equal probability). I therefore do not understand that the ANN can learn something that mathematically is not supposed to exist. And so it is fair to ask if the disorder modelling is sufficiently generic in this paper: it is written that $\delta t$ and $\delta {\mu}$ are constant and isotropic, ie they do not make a further distinction between symmetric and nematic phases. If one makes these distributions broad and anisotropic, do the conclusions of this work still hold?

(3) on p8 the authors write: "In comparison, machine learning approaches are base upon the original real-space LDOS data and thus may look beyond merely the Fermi wave vectors." (note the typo: base --> based). But what do the authors think that distinguishes the two cases? Looking at Fig 9d (please add the color scale to fig 9c), Fig 7d, and Fig 6d, it seems that there is a difference between the x and y direction for k-values in the range 2 to 3. Is this true, and if so, can it be explained?

---

## Round 2 · Author Response

Dear Editor,

Attached please find our manuscript "Detecting nematic order in STM/STS data with artificial intelligence". We are sorry for the delay in responding to the referee comments. We were confused by the latest statement from you on the main page of our manuscript "With only this short report I cannot formulate a recomendation for this manuscript, therefore I am oblished to contact additional referees. I'm sorry if this takes more time." But it seems this statement is out of date and that you have been waiting for our response to the referee who posted a report for some time now. We include this response here and apologize for the delay.

The referee requested both a thorough comparison with experimental data, an explanation for why the performance of the artificial neural network (ANN) on trained data was poor and that we correct a typo.

On the comparison with experimental data, we opted not to compare with additional experiments but instead to argue that the experimental data we did analyze was robustly predicted to be in the nematic phase by the ANN. We now added a subfigure to our report on this analysis which shows the distribution of results from 1000 randomly extracted 16x16 images from the experimental data which cover both different locations on the surface of the material and sampling at different length scales. We believe the results are now convincing that the ANN robustly identifies nematic order in STM data.

As for the poor performance of the ANN on trained data, we believe the referee misinterpreted what we were trying to say in the first version of the manuscript. The ANN performed very well on trained data (we now show a distribution of the ANN predictions on about 10000 images in Fig. 2). What we were trying to do in the first manuscript was to show a case where it didn't do well but still made the correct prediction to demonstrate the hard cases. We search through 10000 images for a few where it performed poorly to do this. In the new version of the manuscript, we dropped this discussion both to avoid any misunderstanding and because isolated examples are not as meaningful as a statistical analysis of how the ANN performs.

Lastly, we have corrected the typo and made many other minor changes throughout the manuscript. Below is our specific response to the referee's report. We hope this new submission can be published without any further delay.

Sincerely, Jeremy B. Goetz Yi Zhang Michael J. Lawler

The manuscript by Goetz et al. reports on detecting subtle nematic order from the local density of states (LDOS) data measured by scanning tunneling microscopy (STM) using supervised machine learning. They simulate LDOS data using various tight-binding methods to train and test their artificial neural network (ANN). They obtain 95% accuracy score with an ANN with a single hidden layer. They test the ANN on a real STM data that is observed to be anisotropic in previous studies. The ANN is able to identify the nematic order with 99% confidence.

The manuscript is well written, the subject is novel and the results are compelling in my opinion.

We thank this referee for his/her encouraging comments and suggestions.

However, the authors test the ANN only on a single anisotropic STM image. I would like to see its performance on a few other isotropic as well as anisotropic STM images. The fact that the ANN performs poorly on simulated data (only %65 Fig. 2) which is trained on compared to the real STM data (%99) is a bit controversial.

We agree with the referee. In the revised manuscript, we show the statistics of neuron outputs across a large number of input samples for both the simulated and the experimental data, instead of referring to the output of a single data sample. Also, the authors have cross-checked their results for consistency. We hope such visualization of statistics offer more clarity and persuasive power for our claims and conclusions.

Typo: For example, inhomogeneous behavior found in strongly correlated materials [can lead] to ...

We thank this referee for pointing this out. We have fixed this typo and other typos, as we carefully revised through the entire manuscript.

---

## Round 2 · List of Changes

1. We add the histograms in Fig. 2, Fig. 3 and Fig. 4 to show data statistics.

  2. We correct typos and grammar across the manuscript.

  3. We have changed the section and subsection labels for consistency with the catalog at the end of Sec. I.

  4. We have revised Appendix C to make the presentation more succinct.

---

## Round 3 · Referee Report · Anonymous (Referee 2) · 2020-5-18

Report

The authors answered to my questions in a satisfactory way. I see no further reason to uphold this paper.

---

## Round 3 · Author Response

We thank this referee for his/her suggestions and comments. We have revised the manuscript accordingly and responded as follows:

(1) Upon coarse-graining the pixels, the authors write that the confidence of the ANN gradually wanes. Is this also seen in the data from the non-interacting model? Or does one have to include disorder that changes over a scale of a dozen lattice sites to reproduce this?

Following this referee's question, we have tested and indeed observed a similar behavior of waning confidence in the non-interacting-model data as the coarse-graining length scale starts to deviate significantly from the training set. We have included the related ANN results in the Appendix and added a sentence in connection to the observation in Fig.4.

(2) One of the main conclusions of this work is that the ANN remains successful in distinguishing nematic from symmetric samples even when standard approaches fail, say in case of strong disorder. This is remarkable. Imagine a sample with very strong disorder. Then, and this is a typical argument for disordered systems, it is impossible to say whether this sample was generated from phase A or from phase B in case the disorder distribution is generic (in which case one can continuously change the parameters of the disorder distribution, and hence generate the sample from either phase with equal probability). I, therefore, do not understand that the ANN can learn something that mathematically is not supposed to exist. And so it is fair to ask if the disorder modeling is sufficiently generic in this paper: it is written that δtδt and δμδμ are constant and isotropic, i.e., they do not make a further distinction between symmetric and nematic phases. If one makes these distributions broad and anisotropic, do the conclusions of this work still hold?

Indeed, in the ultra-strong-disorder limit, there is no physical distinction whatsoever that exists between the phases with different symmetries, and the ANN will not help. On the other hand, the distinctions may remain with the disorder strength below a certain threshold, though the information is likely blurred and hidden, such as the example in Figs. 7 and 8 - averaging over a large number of configurations reveals the existing yet hidden information. Such data is where the machine-learning-based method shines.

In the revised manuscript, we have added results in the Appendix on ANN outputs in the presence of even stronger disorders, where the distinctions between the phases indeed vanish. We have also included a discussion on the generality of the proposed AI perspective in the conclusive remarks.

Also, we did not include anisotropy to the disorder distributions since it makes it difficult to attribute the origin of the anisotropy and distinguish the underlying symmetry of the pristine model labeling the physical phases, which is our original and essential target.

(3) On p8 the authors write: "In comparison, machine learning approaches are base upon the original real-space LDOS data and thus may look beyond merely the Fermi wave vectors." (note the typo: base --> based). But what do the authors think that distinguishes the two cases? Looking at Fig 9d (please add the color scale to fig 9c), Fig 7d, and Fig 6d, it seems that there is a difference between the x and y direction for k-values in the range 2 to 3. Is this true, and if so, can it be explained? -- this amounts to the fact that the ANN output is difficult to interpret for humans.

We thank this referee for pointing out our typo. We have corrected the typo and added a color scale to Fig. 9c. We also agree with the referee on the existence of various differences between the x and y directions.

However, it is still hard to formulate a universal clue that applies more generally, especially since the isotropic cases (e.g., Fig. 6c and 7c) have, in principle, differences between the two directions as well due to the randomness of the quenched disorders. Also, although the peaks are relatively well-defined features in Fig. 6d (Friedel oscillations are present), they are much less apparent features overall in Fig. 7d due to limited resolution and cleanness. Further, Fig. 9d brings even further complication: the peaks lie in nearly identical locations due to similar k_F in both directions, yet the anisotropy is in v_F instead.

Honestly, we do not yet have a solid grasp on what the ANN uses to distinguish the two phases. Understanding how ANN works is an evolving frontier, and at this moment, there lacks a well-controlled method to interpret the ANN analytically in general.

(4) It is not entirely clear to me whether the modeling by a non-interacting model is sufficient in case no Bragg peaks or Friedel oscillations are observed.

We agree with this referee that, ideally, the training should consist of more generic models, including various interacting models and non-interacting models with broader settings for more rigorous and practical applications, which is beyond the scope of the current work. Instead, we focus on simple illustrations that the ANN indeed offers a useful perspective in certain situations, e.g., no Bragg peaks or Friedel oscillations, which were previously found rather difficult.

We have included a discussion on generality and future roadmap in the last section.

---

## Round 3 · List of Changes

• We have included the new ANN results on scaling in the Appendix and added a sentence in connection to the observation in Fig.4.
  • We added new ANN results on disorder effects in the Appendix and a discussion on the generality of the proposed AI perspective in the conclusive remarks.
  • corrected the typo: base-->based on p8 and added a color scale to Fig. 9c.
  • We have included a discussion on generality of our approach and future roadmap in the last section.

---

## Editorial Decision

published